# Patient with Secondary Amyloidosis Due to Crohn’s Disease on Hemodialysis Effectively Treated with Ferric Carboxymaltose Injections: A Case Report and Literature Review

**DOI:** 10.3390/diseases13040125

**Published:** 2025-04-21

**Authors:** Masayo Ueno, Fumihito Hirai, Asami Fuji, Yuko Shimomura, Keiko Uemoto, Kosuke Masutani, Takao Saito

**Affiliations:** 1Department of Blood Purification, Sanko Clinic, 4-9-3 Ropponmatsu, Chuo-ku, Fukuoka 810-0044, Japan; 2Department of Gastroenterology and Medicine, Faculty of Medicine, Fukuoka University, 7-45-1 Nanakuma, Jonan-ku, Fukuoka 814-0180, Japan; 3Division of Nephrology and Rheumatology, Faculty of Medicine, Fukuoka University, 7-45-1 Nanakuma, Jonan-ku, Fukuoka 814-0180, Japan

**Keywords:** ferric carboxymaltose, hemodialysis, iron deficiency anemia, Crohn’s disease

## Abstract

**Background:** Almost all patients undergoing dialysis develop renal anemia and receive medicines such as erythropoietin and iron preparations. However, the conventional intravenous treatment with saccharated ferric oxide (SFO) is insufficient for these patients when they have incurable and persistent iron deficiency anemia due to secondary amyloidosis. Therefore, we administered 500 mg of ferric carboxymaltose (FCM) to such a patient with Crohn’s disease. **Case presentation:** A 56-year-old man on maintenance hemodialysis had secondary amyloidosis due to Crohn’s disease. Additionally, he was anemic and received 40 mg of SFO weekly; however, his hemoglobin (Hb) level remained low at 7 g/dL. Therefore, 500 mg of FCM was administered bimonthly from the first to the fourth dose, and the Hb level temporarily increased compared to that after the previous SFO administration. Since bimonthly administration did not adequately maintain the Hb level, FCM was administered monthly from the 5th to 12th dose, which stabilized the Hb level at 10–12 g/dL. No side effects, such as hypophosphatemia, were observed. **Conclusions:** A single dose of 500 mg FCM administered once every 1–2 months stabilizes the Hb level and contributes to efficient iron utilization in patients with incurable anemia undergoing hemodialysis.

## 1. Background

Although the proportion of patients with secondary amyloidosis undergoing maintenance hemodialysis is low, their prognosis is generally poor due to frequent occurrence of multiorgan damage including renal failure [1,2]. Secondary amyloidosis often develops after chronic inflammatory diseases such as rheumatoid arthritis and tuberculosis; however, Crohn’s disease is another important cause [3,4,5]. Inflammatory bowel diseases such as Crohn’s disease tend to cause iron-deficiency anemia, and such patients, with chronic kidney diseases, should be treated with intravenous iron preparations and erythropoietin [6].

Here, we report the case of a patient with secondary amyloidosis due to Crohn’s disease who was undergoing maintenance hemodialysis. The patient received 40 mg of SFO intravenously for anemia every week. However, since his Hb level gradually decreased, he received intravenous injections of FCM, which dramatically improved his condition as his Hb level increased.

## 2. Case Presentation

### 2.1. Patient

A 56-year-old Japanese man was diagnosed with Crohn’s disease at the age of 22 and was undergoing follow-ups with his family doctor. However, at 39 years, he had worsening abdominal pain, and he was diagnosed with amyloidosis based on a biopsy of the small intestinal mucosa. In the same year, amyloidosis led to renal dysfunction. At 41 years, he experienced anal bleeding and developed end-stage renal disease requiring hemodialysis. Since the age of 53 years, he has presented intermittent melena. Additionally, despite continuously receiving weekly intravenous injections of darbepoetin at 60 µg and SFO at 40 mg, his anemia worsened to a Hb level of 7.1 g/dL. Moreover, there was a decrease in the serum iron level (18 µg/dL), TSAT (4.9%), and serum ferritin level (23.2 ng/mL). Blood counts and biochemical test results are shown in Table 1.

After the patient started dialysis, he underwent monthly chest X-rays and electrocardiograms and annual echocardiograms. However, the cardiothoraic ratio was less than 50%, the electrocardiogram did not reveal cardiomegaly, and there was no evidence of arrhythmia or ischemic heart disease. Furthermore, the echocardiogram did not reveal any valvular heart disease or wall motion abnormalities.

### 2.2. Clinical Course 

Given the low levels of all anemia-related blood parameters and severe iron-deficiency anemia, blood transfusions were administered several times. But the effects were temporary, and it was deemed difficult to continue the transfusion because irregular antibodies were detected. Finally, intravenous injections of 500 mg of FCM were initiated at the Department of Gastroenterology, Fukuoka University Hospital, while hemodialysis was performed three times a week at the Sanko Clinic (Figure 1). Follow-up blood tests after the first administration of FCM showed that the Hb level gradually increased from 7 g/dL to 9 g/dL, which temporarily exceeded 10 g/dL after the second FCM administration. However, when the FCM administration interval was increased to ≥8 weeks, the TSAT remained low at 10% until the 24th week before the 4th dose of FCM, the serum ferritin levels fluctuated greatly and were unstable, and the Hb level reduced to approximately 8 g/dL. Accordingly, the FCM administration interval was shortened to 4 weeks, and the Hb level increased again. After the 13th administration, the Hb level was high at 13.6 g/dL; therefore, the FCM withdrawal period was 20 weeks, and the darbepoetin dose could be reduced from 60 to 40 µg/week. Additionally, the serum ferritin level increased to approximately 100 ng/mL and never fell below 30 ng/mL; TSAT exceeded 20% and decreased after 17 weeks. After the 14th dose, favorable Hb levels were achieved with regular FCM administration. Unfortunately, the patient developed severe COVID-19 complicated by pneumonia and died 15 days after the 18th administration of FCM.

Before FCM administration, SFO was intravenously injected once per week. Therefore, using the weekly average Hb value with this administration method as a control, Figure 2 shows the changes in weekly average Hb values for bimonthly (doses 1–4) and monthly FCM administrations (doses 5–12). When FCM was administered bimonthly (56 days on average), the Hb level gradually increased from the 2nd week onwards compared to when SFO was administered, with a downward trend being observed from the 4th week onwards. Contrastingly, when FCM was administered monthly (31 days on average), the Hb levels remained stable at 10–12 g/dL each week. Serum ferritin levels temporarily increased by >300 ng/mL one week after FCM administration, but the high value did not persist. Furthermore, serum phosphorus levels did not decrease with FCM administration, liver function test values were within the normal range, and side effects such as headaches were not observed.

## 3. Discussion

Regardless of the cause, secondary amyloidosis is often characterized by cardiac amyloidosis and must be differentiated from hypertrophic cardiomyopathy [7,8] and hypertensive heart disease [9]. However, in this case, as described in the case presentation, there were no cardiovascular abnormalities, and although the patient ultimately died of pneumonia due to COVID-19 infection, cardiovascular disorders were not the main cause of death. On the other hand, gastrointestinal amyloidosis due to Crohn’s disease induced refractory and inflammatory intestinal bleeding, and as a result, iron deficiency anemia was persistent and severe.

In dialysis patients, hematopoietic function is also reduced as a result of renal failure, and to improve such renal anemia, it has been common to administer ESAs intravenously, and recently oral administration of HIF-PHIs has also been put to practical use. On the other hand, dialysis patients are often iron deficient due to blood loss into dialysis lines and filters, frequent laboratory testing, and gastrointestinal bleeding, and they lose an average of 1 to 2 g of iron per year [10,11].

Therefore, for ESAs or HIF-PHIs to be maximally effective, it is essential to maintain an iron supply in amounts necessary for Hb synthesis and to compensate for losses. Oral iron administration is not prescribed for the treatment of iron deficiency in patients on dialysis because of incomplete effect, and the KDIGO practice guideline for anemia in chronic kidney disease (CKD) suggests using intravenous iron rather than oral iron. With iron gluconate and iron sucrose, the maximum single dose is usually 125 mg or 200 mg, for a total of 1000 mg per consecutive hemodialysis treatment, and a recent study recommend the 1500mg total cumulative dose of FCM [12]. In Japan, it is recommended that intravenous iron be slowly administered at 40 mg to patients undergoing hemodialysis at the end of a dialysis session once per week by setting 13 administrations as one cycle while confirming improvement in anemia and evaluating the iron status to ensure that serum ferritin levels are <300 ng/mL [13]. According to this guideline, we administered SFO as gastrointestinal bleeding due to Crohn’s disease continued, however, iron stores in the body tended to be insufficient, and Hb was always less than 8 g/dL. In cases of severe anemia, blood transfusion is an option, but in the case of dialysis patients, it is likely to cause hemolytic transfusion reactions, risk of infection, fluid and iron overload, and hyperkalemia, and the improvement of anemia is only temporary [14,15]. Therefore, the use of blood transfusions was limited.

Recently, a 500 mg intravenous formulation of FCM was launched for severe iron deficiency anemia. FCM is a type I polynuclear iron (III) hydroxide carbohydrate complex designed to mimic physiologically occurring ferritin proteins [16]. Serum iron and ferritin reach their maximum blood concentrations 15–30 min and 48–120 h after FCM administration, respectively. Thereafter, total serum iron levels fall below the limit of quantification in most of the patients within 60–96 h, and the urinary excretion rate of FCM is negligible [16].

FCM has been widely used in gynecological diseases and inflammatory bowel diseases that cause iron deficiency anemia; however, there are no recommendations regarding the use of FCM in patients undergoing dialysis in the guidelines of renal anemia by the KDIGO Conference [17] or the Japan Society for Dialysis Therapy [13]. Further, there have been no reported cases regarding the use of FCM in the United States and Japan.

Contrastingly, numerous clinical studies in Europe have demonstrated the efficacy and safety of treatment with FCM (Table 2). Covic et al. [18] conducted a multicenter open-label clinical study on the use of FCM in 163 patients undergoing hemodialysis. They found that intravenously injecting 100–200 mg of FCM two or three times a week for 6 weeks increased the Hb level from 9.1 ± 1.30 g/dL to 10.3 ± 1.63 g/dL in 100 patients, with serious side effects in 7.4%. Hofman et al. [19] reported that, in 221 stable patients undergoing hemodialysis, switching from iron sucrose to FCM at a 1:1 ratio improved the iron status parameters, reduced the dose of ESAs, and increased Hb levels. A similar efficacy was reported after switching from ferric gluconate to FCM, which resulted in a reduction in the administered iron doses [20,21,22].

Therefore, in our presented case, FCM was considered the only way to improve refractory iron deficiency anemia and thus was administered at a dose of 500 mg per administration. Even in Europe, the dose for patients undergoing dialysis is usually ≤200 mg per administration. However, Portolés-Pérezi et al. [23] demonstrated that iron indicators were safely improved when 500 mg of FCM was intravenously infused at once in patients undergoing home dialysis. Diebold and Kistler [24] concluded that intravenous FCM infusion resulted in dose-dependent ferritin elevation of extended duration. Further, they recommended temporal coordination of blood sampling for iron status evaluation and maintaining an intravenous iron dosing schedule. Accordingly, we considered that FCM administration could be used to improve low Hb levels while ensuring safety by performing appropriate evaluations even during hemodialysis, with consideration of the administration interval.

Secondary amyloidosis is associated with prolonged chronic inflammation in patients with Crohn’s disease, which results in the excessive production of various cytokines. Chronically increased levels of cytokines, such as IL-6, result in increased hepcidin production in the liver, which in turn induces ACD by suppressing iron absorption from the gastrointestinal tract and iron recycling via macrophages [25,26]. Therefore, in this case, it is possible that ACD was a contributing factor to the severe anemia rather than just simple renal or hemorrhagic anemia.

A stable Hb level was obtained while gradually reducing the darbepoetin dose by administering 500 mg of FCM once every 1–2 months. Additionally, hypophosphatemia is a known side effect of FCM [27]; however, the patient did not show low serum phosphorus levels or other side effects.

The patient developed COVID-19 two weeks after the 18th administration of FCM and died of respiratory failure due to pneumonia. The relationship between the exacerbation of bacterial or virus infection and iron overload status is known. However, laboratory findings on day 7 of the 18th FCM administration showed a Hb level of 9.9 g/dL; serum iron level, 77 μg/dL; serum ferritin level, 348 ng/mL; and TSAT, 31.3%. Therefore, FCM administration may not play a causal role in COVID-19 infection.

## 4. Conclusions

The guidelines for renal anemia in patients with CKD recommend administering intravenous iron to patients undergoing dialysis at a dose of 40 mg per weekly to avoid iron overload. However, in patients with incurable persistent iron deficiency anemia, such as those with secondary amyloidosis due to Crohn’s disease, a single dose of 500 mg FCM once every 1–2 months may stabilize Hb and contribute towards efficient iron utilization.

However, the limitation of the current study is the case report. In particular, various factors with inflammatory systemic diseases can act as biases. Further trials administrating FCM for many hemodialysis patients with secondary amyloidosis should help strengthen our findings.

## Figures and Tables

**Figure 1 diseases-13-00125-f001:**
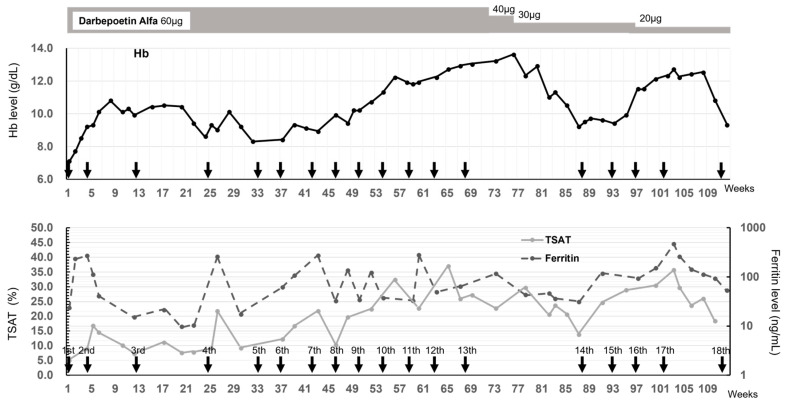
**Time course of Hb level, TSAT, and ferritin level after FCM administration**. The 18 administrations of FCM are indicated by arrows. The change in Hb is indicated by the solid line in the upper figure. The changes in TSAT and ferritin are indicated by the solid and dashed lines in the lower figure, respectively.

**Figure 2 diseases-13-00125-f002:**
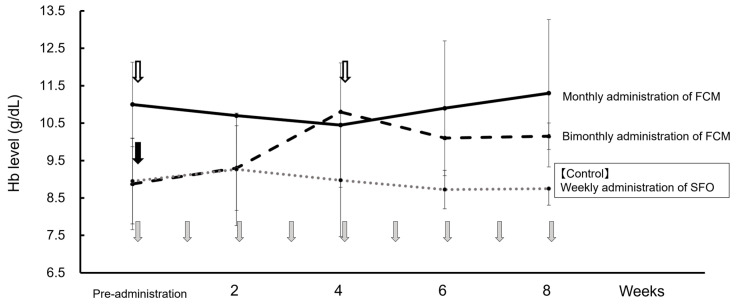
**Changes in the average Hb level at dosing intervals of FCM and SFO**. First to fourth: bimonthly administration of FCM (
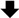
 dashed line), fifth–twelfth: monthly administration of FCM (
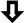
 solid line), and weekly administration of SFO as a control drug before FCM administration (
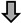
 dotted line). Data are shown as mean ± standard.

**Table 1 diseases-13-00125-t001:** Results of blood examination before the administration of FCM.

Blood Counts				Biochemistry						
WBC	9100	/μL		TP	5.7	g/dL		Na	141	mmol/L
RBC	261 × 10^4^	/μL		Alb	3.1	g/dL		K	5.2	mmol/L
Hb	7.1	g/dL		BUN	78.3	mg/dL		Cl	106	mmol/L
Ht	24.2	%		Cr	14.22	mg/dL		Ca	8.6	mg/dL
MCV	92.7	fL		UA	10.6	mg/dL		IP	5.8	mg/dL
MCH	27.2	pg		AST	24	U/L		Mg	2.4	mg/dL
MCHC	29.3	g/dL		ALT	25	U/L		CRP	0.115	mg/dL
Reticulocytes	30	‰		ALP	59	U/L		Fe	18	μg/dL
Plt	26.6 × 10^4^	/μL		LDH	190	U/L		TIBC	364	μg/dL
				γGTP	14	U/L		TSAT	4.9	%
				TC	134	mg/dL		Ferritin	23.2	ng/mL

WBC, white blood cells; RBC, red blood cells; Hb, hemoglobin; Ht, hematocrit; MCV, mean corpuscular volume; MCH, mean corpuscular hemoglobin; MCHC, mean corpuscular hemoglobin concentration; Plt, platelet.

**Table 2 diseases-13-00125-t002:** Studies of FCM treatments in adult dialysis patients.

Authors /Country ^(a)^	Design	Patients (Number)	Treatment	FCM AP	Results
Covic et al., [18]/Romania	Multicenter, open-label, single-arm, Phase II study	HD (163)	FCM 100 or 200 mg, two to three sessions per week for a maximum of 6 W Hb ≤ 11.0 g/dL and either ferritin ≤200 µg/L or TSAT < 20%	Maximum 6 W	Mean Hb levels increased at 2 W after the first administration and continued to increase through the observation period at follow-up. Within the 2 W following the first administration FCM, ferritin levels and TSAT had increased until the final follow-up visit. 3.1% patients discontinued study medication due to an AE.
Hofman et al., [19] /Netherlands	Retrospective study	HD (221)	Treated with IS 100 mg for 6 M and after washout period with FCM 100 mg for 9 M according to the following criteria. The shift IS to FCM, IS 100 mg or FCM 100 mg TSAT < 20% or Ferritin < 200 µg/L, every week TSAT 20–30% and/or Ferritin 200–500 µg/L, every 2 W TSAT 30–50% and/or Ferritin 500–800 µg/L, every 4 W TSAT > 50% and/or Ferritin > 800 µg/L, no administration	9 M	The dosage of iron medication decreased significantly after switch from IS to FCM (*p* = 0.04), Hb (*p* < 0.001) and hematocrit (*p* < 0.001) increased significantly. After the switch from IS to FCM, ferritin increased significantly (*p* < 0.001) as well as TSAT (*p* < 0.001).
Lacquaniti et al., [20]/Italy	Retrospective study	HD ^(b)^ (25)	Treated with FG 125 mg for 2Y and after washout period with FCM 100 mg for 2 Y according to the following criteria. TSAT < 20% and/or Ferritin < 200 µg, every week TSAT 20–30% and/or Ferritin 200–500 µg, every 2 W TSAT 30–50% and/or Ferritin 500–800 µg, every month	2 Y	FCM increased TSAT levels by 11.9% (*p* < 0.001) with respect to FG. Events of TSAT less than 20% were reduced during FCM. The monthly dose of EPO was reduced in the FCM period (*p* = 0.004). During the period with FCM, ferritin levels were higher than during FG (*p* < 0.001), while transferrin was reduced (*p* = 0.001).
Gobbi et al., [21]/Italy	Single-center, open-label, uncontrolled, prospective study	HD ^(b)^ (24)	FG 125 mg once a week was switched to FCM administration protocol as follows. Ferritin < 100 µg/L, once a week 200 mg Ferritin 100–250 µg/L, once a week 100 mg Ferritin 250–500 µg/L, every 2 W 100 mg Ferritin > 500 µg/L, no administration	12 M	At FCM protocol, ferritin, TSAT, and Hb levels significantly increased (*p* = 0.001). Mean EPO consumption significantly decreased (*p* = 0.001). No patient needed RBC transfusions during the follow-up. No gastrointestinal or other AEs were reported.
Rosati et al., [22]/Italy	Retrospective multicenter observational study	HD ^(b)^ (77)	FG treatment was switched to FCM treatment, when anemic HD patients were unresponsive to FG treatment. The mean dose for the entire period of FG treatment was 394 ± 203 mg compared to 412 ± 243 mg in the period under FCM treatment with no statistical difference between them.	6 M	ERI decreased from 24.2 ± 14.6 at pre-switch to 20.4 ± 14.6 after FCM and Hb levels ≥10.5 g/dL improved from 61% to 75.3% patients (*p* = 0.042). A 1 g/dL increase in Hb levels was seen in 26% of patients as well as a 37.7% increase in patients achieving >20% increase in TSAT after FCM. Levels of Hb, TSAT and ferritin increased during FCM treatment with a concomitant decrease in ESAs. No hypersensitivity reaction was recorded and only one patient reported an AE after FCM. FCM treatment was associated with a cost saving.
Portolés-Pérezi et al., [23]/Spain	Multicenter, retrospective study	PD (91)	FCM 500 or 1000 mg, TSAT < 20% or Ferritin < 100 µg/L TSAT > 20% and Ferritin 200–800 µg/L Ferritin > 800 µg/L, no administration	12 M	68.6% of patients achieved ferritin levels of 200–800 ng/mL, 78.4% achieved TSAT > 20%, and 62.8% achieved TSAT > 20% and ferritin > 200 ng/mL after 12 M of FCM initiation (*p* < 0.01). Hb levels were maintained at >11 g/dL with a lower dose of darbepoetin throughout the follow-up. No hypersensitivity reaction, FCM discontinuation or dose adjustment due to a serious AE was registered.
Diebold and Kistler [24] /Switzerland	Prospective observational study	HD ^(b)^ (39)	FCM 100 or 200 mg, every 4 W Defined by ESAs dose adjustments of <25% within the last 2 M. Hb values between 95 g/L and 125 g/L within the last 12 W with a difference between the lowest and the highest value of <15 g/L	2 M or more	Ferritin values increased by 113 ± 72.2 µg/L (*p* < 0.001) from baseline to the peak value and remained significantly elevated until 2 W after the administration of 100 mg FCM. After the administration of 200 mg FCM, ferritin values increased by 188.5 ± 67.56 µg/L (*p* < 0.001) and remained significantly elevated by the end of week 3. TSAT values increased by 12.0 ± 9.7% (*p* < 0.001) and 23.1 ± 20.4% (*p* = 0.002) in patients receiving FCM, respectively, and returned to baseline within 4 D.

AP: administration period, Y: years, M: months, W: weeks, D: days, HD: hemodialysis, PD: peritoneal dialysis, FCM: ferric carboxymaltose, FG: ferric gluconate, IS: iron sucrose, Hb: hemoglobin, TSAT: transferrin saturation, ERI: erythropoietin resistance index, ESA: erythropoiesis stimulating agent, EPO: erythropoietin, RBC: red blood cell, AE: adverse event. ^(a)^ Country of main research facility, ^(b)^ patient treated with FG in the past for ≥6 months.

## Data Availability

The data presented in this study are available to request to the corresponding author.

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
