# Peer review of "Patient with Secondary Amyloidosis Due to Crohn’s Disease on Hemodialysis Effectively Treated with Ferric Carboxymaltose Injections: A Case Report and Literature Review"

_diseases, 2025, doi:10.3390/diseases13040125_

Round 1

Reviewer 1 Report

Comments and Suggestions for Authors

Amyloidosis can present with cardiac hypertrophy, often mimicking various cardiac and non-cardiac conditions. A multidisciplinary evaluation is essential for accurate diagnosis and optimal management, as outlined in this expert document PMID: 31983465, which should be included in the reference list.

Comments on the Quality of English Language

The quality of scientific English is modest; shorter, more concise periods should be preferred.
A complete revision of the article is recommended in preference to more descriptive clarity.

Author Response

Amyloidosis can present with cardiac hypertrophy, often mimicking various cardiac and non-cardiac conditions. A multidisciplinary evaluation is essential for accurate diagnosis and optimal management, as outlined in this expert document PMID: 31983465, which should be included in the reference list.

Reviewer 2 Report

Comments and Suggestions for Authors

Authors present a clinical case of a patient with Crohns disease with an administration of 500 mg of ferric carboxymaltose. A single dose of 500 mg FCM administered once every 1–2 months stabilizes the Hb. That being said, the presentation  of the case is good.

But the discussion of the literature should be improved:

  • The discussion focuses on FCM treatment; it should explore some alternative therapies and their weakness
  • What might be reasons why conventional iron supplementation failed?
  • Authors should mention limitation of cited studies
  • Please format your graphs to have clear legends
  • I think, due to some grammar issues, a thorough proofreading for language consistency is needed.
  • Please add limitations including possible biases of the presented case
Comments on the Quality of English Language

Grammar errors have to be corrected

Author Response

Authors present a clinical case of a patient with Crohns disease with an administration of 500 mg of ferric carboxymaltose. A single dose of 500 mg FCM administered once every 1–2 months stabilizes the Hb. That being said, the presentation  of the case is good.

But the discussion of the literature should be improved:

  • The discussion focuses on FCM treatment; it should explore some alternative therapies and their weakness
  • What might be reasons why conventional iron supplementation failed?
  • Authors should mention limitation of cited studies
  • Please format your graphs to have clear legends
  • I think, due to some grammar issues, a thorough proofreading for language consistency is needed.
  • Please add limitations including possible biases of the presented case

Reviewer 3 Report

Comments and Suggestions for Authors

I would congratulate with authors for the very good paper and case. Secondary amyloidosis is associated with prolonged chronic inflammation in patients with Crohn disease; results are interesting since in a patient with incurable persistent iron deficiency anemia, such as those with secondary amyloidosis due to Crohn disease, a single dose of 500 mg FCM once every 1– 2 months may stabilize Hb and contribute towards efficient iron utilization. I have only one minor comment: authors in discussion should more discuss on potential cardiac involvement that is a fundamental issue and totally missing. In particular authors should more focus on differential diagnoses from secondary cardiac amyloidosis such as a pre-existent hypertrophic cardiomyopathy (DOI: 10.14740/cr496w), athlete’s heart (DOI: 10.1111/jce.14526) and hypertensive heart disease (DOI: 10.1016/j.medcli.2023.10.006). Please expand the important discussion since this scenario may impact also on treatment and cite 3 suggested references

Author Response

I would congratulate with authors for the very good paper and case. Secondary amyloidosis is associated with prolonged chronic inflammation in patients with Crohn disease; results are interesting since in a patient with incurable persistent iron deficiency anemia, such as those with secondary amyloidosis due to Crohn disease, a single dose of 500 mg FCM once every 1– 2 months may stabilize Hb and contribute towards efficient iron utilization. I have only one minor comment: authors in discussion should more discuss on potential cardiac involvement that is a fundamental issue and totally missing. In particular authors should more focus on differential diagnoses from secondary cardiac amyloidosis such as a pre-existent hypertrophic cardiomyopathy (DOI: 10.14740/cr496w), athlete’s heart (DOI: 10.1111/jce.14526) and hypertensive heart disease (DOI: 10.1016/j.medcli.2023.10.006). Please expand the important discussion since this scenario may impact also on treatment and cite 3 suggested references

Round 2

Reviewer 2 Report

Comments and Suggestions for Authors

-

Reviewer 3 Report

Comments and Suggestions for Authors

no further comments